# Dietary Patterns Associated with Diabetes in an Older Population from Southern Italy Using an Unsupervised Learning Approach

**DOI:** 10.3390/s22062193

**Published:** 2022-03-11

**Authors:** Rossella Tatoli, Luisa Lampignano, Ilaria Bortone, Rossella Donghia, Fabio Castellana, Roberta Zupo, Sarah Tirelli, Sara De Nucci, Annamaria Sila, Annalidia Natuzzi, Madia Lozupone, Chiara Griseta, Sabrina Sciarra, Simona Aresta, Giovanni De Pergola, Paolo Sorino, Domenico Lofù, Francesco Panza, Tommaso Di Noia, Rodolfo Sardone

**Affiliations:** 1Unit of Data Sciences and Technology Innovation for Population Health, National Institute of Gastroenterology “Saverio de Bellis”, Research Hospital, 70013 Bari, Italy; rossella.tatoli@irccsdebellis.it (R.T.); rossydonghia@irccsdebellis.it (R.D.); fabio.castellana@irccsdebellis.it (F.C.); zuporoberta@gmail.com (R.Z.); sarah.tirelli@irccsdebellis.it (S.T.); sara.denucci@irccsdebellis.it (S.D.N.); annamaria.sila@irccsdebellis.it (A.S.); annalidia.natuzzi@irccsdebellis.it (A.N.); chiara.griseta@irccsdebellis.it (C.G.); sabrinasciarra@hotmail.it (S.S.); arestasimo@gmail.com (S.A.); rodolfo.sardone@irccsdebellis.it (R.S.); 2Neurodegenerative Disease Unit, Department of Basic Medicine, Neuroscience, and Sense Organs, University of Bari Aldo Moro, 11, 70125 Bari, Italy; madia.lozupone@gmail.com (M.L.); f_panza@hotmail.com (F.P.); 3Unit of Geriatrics and Internal Medicine, National Institute of Gastroenterology “Saverio de Bellis”, Research Hospital, 70013 Bari, Italy; gdepergola@libero.it; 4Department of Electrical and Information Engineering, Polytechnic of Bari, 70125 Bari, Italy; paolo.sorino@irccsdebellis.it (P.S.); domenico.lofu@poliba.it (D.L.); tommaso.dinoia@poliba.it (T.D.N.)

**Keywords:** diabetes, older adults, dietary pattern, unsupervised learning approach

## Abstract

Dietary behaviour is a core element in diabetes self-management. There are no remarkable differences between nutritional guidelines for people with type 2 diabetes and healthy eating recommendations for the general public. This study aimed to evaluate dietary differences between subjects with and without diabetes and to describe any emerging dietary patterns characterizing diabetic subjects. In this cross-sectional study conducted on older adults from Southern Italy, eating habits in the “Diabetic” and “Not Diabetic” groups were assessed with FFQ, and dietary patterns were derived using an unsupervised learning algorithm: principal component analysis. Diabetic subjects (*n* = 187) were more likely to be male, slightly older, and with a slightly lower level of education than subjects without diabetes. The diet of diabetic subjects reflected a high-frequency intake of dairy products, eggs, vegetables and greens, fresh fruit and nuts, and olive oil. On the other hand, the consumption of sweets and sugary foods was reduced compared to non-diabetics (23.74 ± 35.81 vs. 16.52 ± 22.87; 11.08 ± 21.85 vs. 7.22 ± 15.96). The subjects without diabetes had a higher consumption of red meat, processed meat, ready-to-eat dishes, alcoholic drinks, and lower vegetable consumption. The present study demonstrated that, in areas around the Mediterranean Sea, older subjects with diabetes had a healthier diet than their non-diabetic counterparts.

## 1. Introduction

Diabetes mellitus (DM) is a group of metabolic diseases characterized by an increase in blood glucose concentrations (hyperglycemia). There are two major subtypes of DM: type 1 and type 2. Type 2 diabetes (T2DM) is the most common type of DM (around 90% of people with diabetes have T2DM) [1]. It is mainly linked to insulin resistance (IR) and relatively poor insulin secretion.

Diabetes has become a major public health concern worldwide due to its growing epidemic prevalence. According to the data of the International Diabetes Federation (IDF), diabetes affected 463 million people between the ages of 20 and 79 worldwide in 2019, which will grow to an estimated 700 million by 2045 [2]. Disease prevalence has doubled in Italy in the last 30 years (now 5.7–6.2%, or one in every six people over 65 years old) [3].

T2DM and its complications constitute a major public health problem worldwide, affecting almost all populations in both developed and developing countries, with high rates of diabetes-related morbidity and mortality [4]. In Italy, diabetes is the leading cause of blindness, the second-leading cause of end-stage renal failure requiring dialysis or transplantation, the leading cause of non-traumatic amputation of the lower limbs, and a contributing cause in 50% of heart attacks and strokes [5].

The quick growth of this “diabetes epidemic” is explained by the increase in obesity and overweight, the spread of wrong eating habits and sedentary lifestyles, and population aging [6].

It is well known that healthy lifestyles contribute to the maintenance of normal body weight and the prevention of T2DM. The increased proportion of diseases attributable to diabetes highlights not only the importance of diabetes prevention but also the importance of proper disease management. Dietary behaviour is a core element of diabetes self-management [7,8], with the aim of achieving a good control of plasma glucose levels and thereby preventing long-term complications [9].

Specifically, nutrition guidelines regard the macronutritional composition of diets [10]. There are no remarkable differences between nutritional guidelines for people with T2DM and healthy eating recommendations for the general public [11]. These recommendations refer to the principles of the Mediterranean Diet (MD) as a beneficial dietary pattern associated with numerous health benefits, in both Mediterranean and non-Mediterranean populations [12,13,14].

Recent data suggest that, generally, the Italian adherence to MD is optimal in middle-aged (50–64 years) and older subjects (≥65 years), particularly those living in Southern Italian regions [15]. The presence of some diseases, such as diabetes, together with other age-related barriers can make it difficult for an aging population to follow a correct, healthy dietary pattern.

A recent systematic review of randomized controlled trials examined the effectiveness of different dietary patterns in managing type 2 diabetes [16]. Several reviews suggested that vegetarian and Mediterranean dietary patterns may be more effective in improving glycemic control and certain cardiovascular risk markers in people with diabetes [16]. By contrast, the evidence for the long-term efficacy of low-carbohydrate diets on individuals with T2DM was inconclusive. However, longer-term intervention studies to support these hypotheses are lacking [16].

In the scientific literature, there are currently no studies evaluating the association between diet and diabetes in an older Italian population using advanced statistical techniques such as machine learning.

This study aimed to evaluate dietary differences between subjects with and without diabetes among non-institutionalized older adults from Southern Italy using an unsupervised machine learning approach in the identification of dietary patterns based on principal component analysis.

## 2. Materials and Methods

### 2.1. Study Population

Study participants were residents of Castellana Grotte, Bari, Southern Italy, and the sampling framework is based on the health registry office list on 31 December 2014. This included 19,675 people, 4021 of whom were aged 65 years or older. They were enrolled in the “Salus in Apulia Study”, a public health initiative financed by the Italian Ministry of Health and the Regional Government of Apulia and carried out by the IRCCS research hospital Saverio De Bellis. Previous prospective MICOL studies [17], which began in 1981, included these same potential research subjects. From 2014 to 2018, all eligible subjects, starting with MICOL participants, were invited to take part in the study [18]. All participants signed informed consent acknowledgements after receiving full information about their medical data to be studied. From the entire sample, only 1399 who underwent dietary assessments and clinical evaluations were included in this analysis.

The study was conducted in accordance with the Helsinki Declaration of 1975. Every examination and informed consent form was approved by the Institutional Review Board of the National Institute of Gastroenterology and Research Hospital. All study information is stored in electronic databases that are protected according to Italian privacy laws.

### 2.2. Dietary Assessment and Clinical Evaluation

Diet and eating habits were assessed with a validated food frequency questionnaire (FFQ) used in previous studies [19,20]. FFQ referred only to the frequency of intake and did not consider differences in portion sizes. This questionnaire investigated dietary habits over the previous year and inquired about the consumption of 85 food items, which were further summarized in 28 food groups [18]. Appendix A shows the concordance of single foods in the questionnaire and the food grouping used in the analyses [21].

The self-administered questionnaire was checked for completeness during an interview conducted by a physician at the study centre. The questionnaire also included questions about lifestyle aspects such as educational level, physical activity, and smoking habits. Additionally, at the interview, anthropometric data on waist circumference (cm), weight (kg), and height (cm) were obtained. Weight and height were measured with the mechanical scale SECA 700 and stadiometer SECA 220 (Seca GmBH and Co., Hamburg, Germany), and the body mass index (BMI) was then derived and calculated as the ratio of weight (kg) to height squared (m^2^). The waist circumference was assessed with respect to to the National Cholesterol Education Program: Adult Treatment Panel III (NCEP: ATP III) criteria. Diabetes mellitus was categorized as fasting blood glucose (FBG) ≥ 126 mg/dL.

### 2.3. Statistical Analysis

Patients’ characteristics were reported as mean ± standard deviation (M ± SD) for continuous variables and as frequencies and percentages (%) for categorical variables. To test associations with diabetes-related diseases between groups, the Chi-square test was used for categorical variables, and the Wilcoxon rank-sum (Mann–Whitney) test was used for continuous variables.

To further reduce the number of the 28 food groups, a dimensionality reduction algorithm based on unsupervised learning was used, namely principal component analysis (PCA).

The PCA algorithm finds linear combinations of raw features (also known as projection) such that they retain as much variation in the data as possible, summarized in as few new variables (components) as possible. The vectors (loadings) describing these linear transformations produce a new set of features called scores (eigenvalues), which are uncorrelated with each other. The principal components returned by statistical software are often ranked in descending order by their corresponding eigenvalues, which simply comprise the amount of variance in the original data explained by each principal component. The PCs with the largest eigenvalues account for most of the variation in the data. We applied PCA to the 28 food intakes in the groups of diabetics and in the control groups of non-diabetics. We considered only the most predominant PC (higher eigenvalues) in both groups, describing the loadings for each food in that PC. Due to the nature of this method, the observed food group contribution to the PCA-derived habitual dietary patterns tended to be higher for large meals with a low consistency of consumption and high interindividual variation. We chose PCA as the basis of our analysis of dietary patterns because it is the exploratory method most frequently adopted [22].

All analyses were performed using StataCorp. 2021. Stata Statistical Software: Release 17. College Station, TX, USA: StataCorp LLC.

## 3. Results

The sample analyzed in the present study included 1399 subjects drawn from the “Salus in Apulia Study” population, with an average age of 73.43 ± 6.30 years old. The male gender was slightly predominant, accounting for 53.6% (*p* = 0.02. It was fairly well balanced for the education level, which averaged 6.79 ± 3.79 years of schooling, as well as for mean BMI, 28.98 ± 4.26, and waist circumference, 103.48 ± 10.25 cm. The population was generally overweight had a greater abdominal circumference value than recommended. The study population was subdivided into two groups based on the presence or absence of diabetic disease: i.e., the “Not Diabetic” and the “Diabetic” groups.

Table 1 shows differences between the two groups regarding socio-demographic, lifestyle, and biochemical parameters.

Table 2 shows the characteristics of the two groups in terms of food group consumption. Diabetic subjects consumed more potatoes (13.31 g ± 16.38 vs. 14.01 g ± 31.18), more ready-to-eat dishes (33.24 g ± 34.83 vs. 34.45 g ± 94.18), fewer eggs (8.33 g ± 9.12 vs. 7.40 g ± 8.64), fewer nuts (7.56 g ± 15.72 vs. 5.49 g ± 16.04), and fewer sugary foods and beverages (sweets: 23.74 g ± 35.81 vs. 16.52 g ± 22.87; sugary foods: 11.08 g ± 21.85 vs. 7.22 g ± 15.96; juices: 6.96 g ± 20.64 vs. 4.80 g ± 21.26) than ND subjects.

PCA was used to evaluate dietary differences between the “Diabetic” and “Not Diabetic” groups.

Figure 1 shows that the most significant PCA in the “Diabetic” group was dominated in terms of loading scores by foods of plant origin. The food pattern in this group not only reflected a high-frequency intake of dairy products, eggs, vegetables and greens, nuts, and olive oil but also sweets and sugary foods. This pattern is named the “Vegetarian Pattern”.

Figure 2 shows the food pattern of the “Not Diabetic” group. It was characterized by a high-frequency intake of red and processed meat, seafood, high calorie drinks, ready-to-eat dishes, wine, beer, and spirits.

## 4. Discussion

This cross-sectional study carried out in a population of 1399 Italian middle-aged participants from Castellana Grotte (Puglia, Italy) described the dietary composition of subjects with and without diabetes and identified a dietary pattern characteristic of diabetics.

The major finding was as follows: Subjects with diabetes showed vegetarian-type dietary patterns compared to subjects without diabetes. In our study sample, the average age of the “Diabetic” group was 75 years old, while non-diabetic subjects were slightly younger, the average age being 73 years. Diabetes is a disease with the highest prevalence among the elderly, due to the lengthening of the average life expectancy of the population. The prevalence of this disease rises with increasing age, reaching a value of about 20% in subjects aged 74 years or older [23]. These epidemiological data support the strong association between age and diabetes.

Regarding biochemical parameters, in addition to glucose, which was expected to be higher in subjects with diabetes, the “Diabetic” group also presented higher values of triglycerides, systolic blood pressure, IL-6, and TNF-α. In several cases, diabetes occurs in association with other diseases such as obesity, hypertriglyceridemia, and arterial hypertension. These are some of the main components of metabolic syndrome [24,25]. Nearly 90% of individuals with T2DM are overweight or obese. Obesity is characterized by high levels of several proinflammatory markers, including IL-6 and TNF-α, that cause chronic low-grade inflammation. This condition may play a role in the pathogenesis of obesity-related metabolic disorders and metabolic syndrome [26,27,28,29,30]. The connection between diabetes and metabolic syndromes is a clear explanation of our results.

In this study, we compared the eating habits of the “Diabetic” group with those of the “Not Diabetic” group. Traditional approaches that investigate diet and disease association are mainly focused on single foods or nutrients. Instead, we created food groups starting from single foods.

In the diabetic subjects, the consumption of fruits, vegetables, and nuts was higher than in the “Not Diabetic” group, while the consumption of red meat, processed meat, and ready-to-eat dishes was lower. Participants in both groups consumed sweets and sugary products, although consumption was higher in those with diabetes.

This result reflects different eating habits between the “Diabetic” and “Not Diabetic” groups, although dietary recommendations for good diabetes management do not differ much from those for a healthy diet among the general population. However, people with diabetes are more likely to undergo nutritional counselling than healthy individuals [11].

According to the recommendations of the Italian Society of Diabetology (SID), fruits, vegetables, legumes, and whole grains should never be lacking in a diabetic diet. They are rich in fiber, micronutrients, and phytochemicals, and they ensure good control of blood sugar, triglycerides, and body weight [31,32]. The consumption of at least five portions of fruits and vegetables per day and four portions of legumes per week should introduce the right amount of fiber for good glycemic control. A study conducted on T2D patients showed that a fiber intake equal to or greater than 40 g per day, half of which is water-soluble, is associated with a 10% reduction in mean blood sugar and a 25% reduction in post-meal glycemia [33]. Similar effects were also obtained with smaller, more acceptable, and practicable quantities.

The positive effects on glycemic response were also observed with the intake of whole grains, such as oats and barley, due to the presence of soluble fiber and smaller bioactive compounds such as phenolic compounds [34].

In addition to ensuring a good intake of fiber, fruits and vegetables are a good source of micronutrients (folate; potassium; magnesium; and vitamins A, C, E, and K) and phytochemicals [35], particularly flavonoids, which may be responsible for several health benefits [36,37,38]. In fact, increasing the consumption of antioxidant-rich fruits and vegetables is recommended [39] due to the preventive effects of some phytochemicals against cancer and cardiovascular disease (CVD) [40]. However, the beneficial health effects are not generally explained by single nutritional components, but rather by their combinations and synergistic interactions [41]. In order to benefit from the protective and preventive effects of fruit and vegetables against several chronic diseases, such as type 2 diabetes, CVD, and different types of cancer, the World Health Organization (WHO) recommends a minimum consumption of 400 g, or five portions of 80 g each, of fruits, greens, and/or vegetables per day [42]. In Europe, although fruits and vegetable products are appreciated by older people [43], their consumption in this part of the population is insufficient and, in several cases, below the portions recommended by WHO [44]. In Italy, the situation is the same. According to the data reported by the National Institute of Statistics (ISTAT), the average consumption of fruits and vegetables of Italians over 65 years of age does not satisfy the quantities suggested by proper nutrition guidelines. Indeed, only 11% started to consume the five portions recommended per day, while 44% consume three servings. These percentages are the same for both genders but they differ by age and birth region; moreover, the intake of fruits and vegetables decreases with age, passing from 59% in subjects between 65 and 74 years of age to 44% in subjects over 85, and is greater in northern regions than in the central or southern ones [45]. Contrary to what emerges from the national data, in both groups of our sample, the portions of fruits and vegetables suggested by guidelines were largely satisfied. We could explain this result by underlining the strong bond of our study population with the food traditions of the Mediterranean territories.

Fruits and vegetables are the basis of the Mediterranean food culture. The Mediterranean Diet food pyramid includes their consumption at every main meal [46]. On the contrary, red and processed meat are among the foods at the top of the pyramid, to be consumed no more than once weekly. Regarding diabetes, Fretts et al. showed that processed meat was associated with higher fasting glucose, while unprocessed red meat was associated with both higher fasting glucose and fasting insulin concentrations [47].

Despite recommendations, in our sample, both the “Diabetic” and “Not Diabetic” groups consumed sweets and sugary products, but the “Diabetic” group subjects did so to a greater extent. These foods are characterized by a high glycemic index and load. For this reason, their consumption should be limited in subjects with and without diabetes. In fact, in both the general population and in diabetics, the intake of “added sugars” should not exceed 10% of their total energy intake. Several studies have demonstrated that people who consume more sweetened beverages and desserts show a higher risk of developing central obesity, IR, and T2DM [48,49,50,51].

### Strengths and Limitations

The present study evaluated dietary differences between subjects with and without diabetes drawn from non-institutionalized older adults from Southern Italy, using an unsupervised machine learning algorithm. The main strength of our study is that no study has yet analyzed these aspects in similar populations using this novel approach. Another strength is the description of a dietary pattern characteristic of diabetic subjects.

However, some limitations must be considered. One of the main limitations of the study is the use of food frequencies instead of calculating quantitative daily intake. This type of measurement could increase the bias that is usually associated with a retrospective dietary assessment over a period of one year when compared to the actual intake. Nevertheless, despite the reported limitation of this assessment method, FFQs remain the dietary assessment method most used to study dietary patterns and population eating habits [45,46,47,48,49,50,51,52]. Another important limitation is the nature of the study, which was cross-sectional and does not allow a clear directionality of an association to be discerned. Moreover, conclusions in this study should be considered a descriptive comparison of the dietary pattern between two groups, without quantitative statistical inferences. Another limitation concerns the age of the diabetic subjects in our sample, which is reduced compared to the age range of the typical diabetic population.

## 5. Conclusions

In conclusion, the diet among diabetic subjects in our study population differed considerably from the diet prevalent among non-diabetic subjects. It featured a rich content of fruits, vegetables, dairy products, eggs, vegetables and greens, nuts, and olive oil and a poor content of red meat and processed meat. Only sweets and sugary products were commonly consumed by both groups; therefore, we can define the dietary pattern that characterized subjects with diabetes as “Vegetarian”. On the contrary, non-diabetic subjects had a higher consumption of red and processed meat, ready-to-eat dishes, high calorie drinks, beer, wine, and spirits. Given the greater likelihood of diabetic individuals undergoing nutritional counselling interventions, these results underline the importance of nutritional education as an efficacious tool in primary and secondary prevention. More attention should also be paid to nutrition in the healthy population.

## Figures and Tables

**Figure 1 sensors-22-02193-f001:**
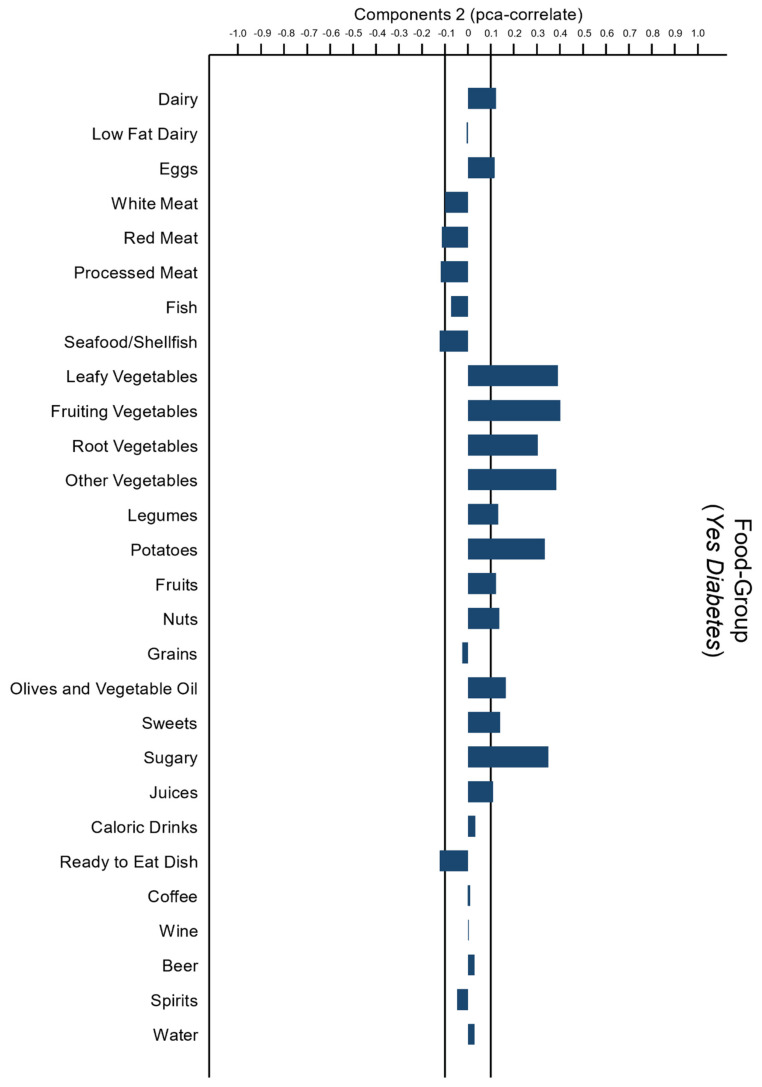
Principal component analysis (PCA) used to identify a dietary pattern of “Diabetic” subjects.

**Figure 2 sensors-22-02193-f002:**
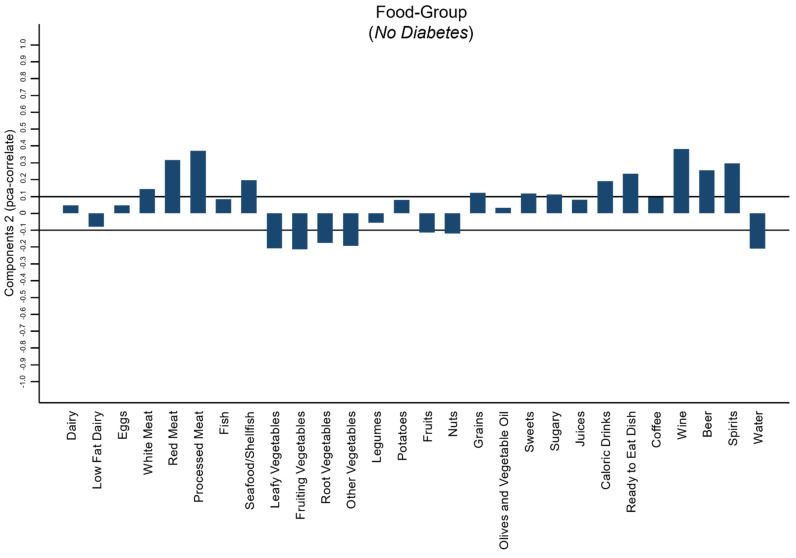
Principal component analysis (PCA) used to identify a dietary pattern of “Not Diabetic” subjects.

**Table 1 sensors-22-02193-t001:** Sociodemographic and clinical variables in patients with and without diabetic disease (Yes/No). The Salus in Apulia Study (*n* = 1399).

Diabetic Disease			
Parameters *	No	Yes	*p* ^ψ^
	(*n* = 1212)	(*n* = 187)	
Gender (%)			0.02 ^
M	634 (52.31)	115 (61.50)	
F	578 (47.69)	72 (38.50)	
Age (yrs)	73.24 ± 6.26	74.66 ± 6.39	0.003
Education (yrs)	7.07 ± 3.80	6.52 ± 3.78	0.05
BMI (kg/m^2^)	28.90 ± 4.34	29.07 ± 4.18	0.60
Normal weight (BMI ≤ 24.90)	221 (18.54)	28 (15.05)	
Overweight (BMI 25.0–29.90)	548 (45.97)	93 (50.00)	
Obese (BMI ≥ 30)	423 (35.49)	65 (34.95)	
Waist (cm)	102.92 ± 10.42	104.05 ± 10.08	0.24
*Biomarkers*			
Glucose (mg/dL)	98.11 ± 11.33	160.63 ± 44.98	<0.0001
Cholesterol (mg/dL)	185.89 ± 36.87	167.47 ± 36.61	<0.0001
HDL (mg/dL)	49.41 ± 13.03	42.95 ± 10.63	<0.0001
LDL (mg/dL)	115.32 ± 31.14	97.78 ± 30.54	<0.0001
Triglycerides (mg/dL)	101.84 ± 54.25	133.58 ± 78.68	<0.0001
Systolic Blood Pressure (mmHg)	132.76 ± 14.30	136.90 ± 14.76	0.0006
Diastolic Blood Pressure (mmHg)	78.48 ± 7.72	77.46 ± 8.19	0.04
IL-6 (pg/mL)	3.85 ± 6.73	4.39 ± 6.48	0.0001
TNF-α (µg/mL)	2.76 ± 3.87	3.16 ± 2.98	0.01

* As mean and standard deviation for continuous and percentage (%) for categorical variables. ^ψ^ Wilcoxon rank-sum test (Mann–Whitney). ^ Chi-square test.

**Table 2 sensors-22-02193-t002:** Food groups average consumption in patients with and without diabetic disease (Yes/No). The Salus in Apulia Study (*n* = 1399).

Diabetic Disease			
Parameters *	No	Yes	*p* ^ψ^
	(*n* = 1212)	(*n* = 187)	
*Food-Groups* ^¥^			
Dairy	104.19 ± 111.15	109.38 ± 99.20	0.41
Low-Fat Dairy	101.84 ± 108.35	98.18 ± 107.52	0.49
Eggs	8.33 ± 9.12	7.40 ± 8.64	0.02
White Meat	26.32 ± 32.52	28.19 ± 59.34	0.82
Red Meat	22.62 ± 23.62	25.99 ± 39.21	0.17
Processed Meat	15.11 ± 15.45	17.57 ± 40.64	0.50
Fish	25.20 ± 23.95	33.64 ± 100.18	0.39
Seafood/Shellfish	9.45 ± 13.75	14.84 ± 64.34	0.31
Leafy Vegetables	59.02 ± 60.42	65.59 ± 93.65	0.94
Fruiting Vegetables	93.39 ± 78.56	107.85 ± 105.38	0.08
Root Vegetables	11.81 ± 26.78	14.17 ± 33.44	0.17
Other Vegetables	80.28 ± 77.02	93.80 ± 106.76	0.28
Legumes	37.78 ± 27.66	41.27 ± 46.99	0.99
Potatoes	13.31 ± 16.38	14.01 ± 31.18	0.002
Fruits	620.23 ± 537.58	598.35 ± 485.11	0.89
Nuts	7.56 ± 15.72	5.49 ± 16.04	<0.0001
Grains	157.59 ± 108.42	145.80 ± 99.22	0.29
Sweets	23.74 ± 35.81	16.52 ± 22.87	<0.0001
Sugary foods	11.08 ± 21.85	7.22 ± 15.96	<0.0001
Juices	6.96 ± 20.64	4.80 ± 21.26	0.002
High Calorie Drinks	7.31 ± 42.37	16.85 ± 95.24	0.53
Ready-to-Eat Dishes	33.24 ± 34.83	34.45 ± 94.18	0.01
Coffee	46.41 ± 29.97	50.32 ± 28.72	0.06
Wine	121.98 ± 162.88	124.38 ± 169.39	0.85
Beer	19.54 ± 73.26	19.56 ± 69.59	0.85
Spirits	1.54 ± 5.48	1.31 ± 5.31	0.62
Water	653.61 ± 297.74	705.75 ± 312.98	0.03

* As mean and standard deviation for continuous and percentage (%) for categorical variables. ^¥^ Food groups were calculated by quantity of daily consumption. ^ψ^ Wilcoxon rank-sum test (Mann–Whitney).

## Data Availability

The datasets used and/or analyzed during the current study are available from the corresponding author upon reasonable request.

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
