# Peer review of "Dietary Patterns Associated with Diabetes in an Older Population from Southern Italy Using an Unsupervised Learning Approach"

_sensors, 2022, doi:10.3390/s22062193_

Round 1

Reviewer 1 Report

This study’s aim was to analyse dietary differences via a FFQ in a cohort of type 2 diabetics using PCA methods. This T2D  group was compared to a group of healthy subjects that are part of a large longitudinal study of older populations living around the Mediterranean.

Overall, this study was well-designed and produced interesting results. However, there are some issues with the rationale and design of the intervention itself that should be addressed further in the manuscript. For example, the diabetic population is drawn from the same cohort as the healthy controls who have an average age of ~74. When the population is this old it's not clear what you are measuring in the T2D population, since it's likely they are very different from the usual population of T2D, since they have survived this long they are likely a group who adopted a more healthy lifestyle including a high quality diet, which helped them to live to this age. Thus this study has a limitation in that it's targeting a specific sub-group of people with T2D (Even the fact they could fill out the questionnaire is indicative of probable better health). It would benefit the paper if more data on the T2D group is added, such as the length of time from initial diagnosis.

Detailed Comments: 

Abstract:

The abstract should also include data on intake of sugary foods, since this is a highly relevant food group to track for diabetes.

Introduction:

The introduction seems to imply that diabetics receive similar healthy eating guidance to the normal population, which seems hard to believe since diabetics are usually told to keep track of carbohydrate and sugar intake to a much higher degree than non-diabetics, especially if they are insulin-dependent.

Methods:

It’s not clear how the study population was chosen out of the original cohort of 19,675 people. Were the subjects selected based on complete FFQ records?

Results:

Table 1: please report the average BMI of the groups, also a break-down of overweight, obese and normal weight.

Table 1: please report the average weight in both groups

Table 2 What are Fruiting Vegatables? What are High Calorie Drinks?

Discussion:

The average age of the diabetic population is referred to as 74 in Table 1 and 75 in the Discussion. Please be consistent. 

Obesity is referred to as a risk factor for diabetes, but we don’t have data on the obesity % of the different groups.

The discussion mentions the recommended servings of fruit and vegetables, it would be helpful to know what amounts of fruits/vegetable the subjects consumed. 

Limitation: It’s likely that this diabetic population is somewhat special from a typical diabetic population with a wider range of ages, and this limitation should be addressed.

Author Response

Firstly, we would like to thank the reviewer for their suggestions and for the precious time spent reviewing our manuscript.

Abstract:

The abstract should also include data on intake of sugary foods since this is a highly relevant food group to track for diabetes.

Thanks for your suggestion. We included the required information in the abstract (see lines 31-32).

Introduction:

The introduction seems to imply that diabetics receive similar healthy eating guidance to the normal population, which seems hard to believe since diabetics are usually told to keep track of carbohydrate and sugar intake to a much higher degree than non-diabetics, especially if they are insulin-dependent.

Thanks for your comment. This statement in the introduction was made exclusively for people with type 2 diabetes. We modified this statement to be more transparent (see line 67). 

Methods:

It’s not clear how the study population was chosen out of the original cohort of 19,675 people. Were the subjects selected based on complete FFQ records?

Thanks for your comment. In this study, we selected subjects that underwent the dietary assessment and clinical evaluation (see lines 102-103).

Results:

Table 1: please report the average BMI of the groups, also a break-down of overweight, obese and normal weight.

We modified Table 1 by adding the required information.

Table 1: please report the average weight in both groups

We modified Table 1 by adding the required information.

Table 2 What are Fruiting Vegetables? What are High Calorie Drinks?

We added a supplementary table with the list of foods that make up each food group (see “STable 1”).

Discussion:

The average age of the diabetic population is referred to as 74 in Table 1 and 75 in the Discussion. Please be consistent.

We modified the value in the discussion (see line 230).  

Obesity is referred to as a risk factor for diabetes, but we don’t have data on the obesity % of the different groups.

Thanks for your comment. Please see Table 1 where information on the number of obese subjects in the 2 groups was added.

The discussion mentions the recommended servings of fruit and vegetables, it would be helpful to know what amounts of fruits/vegetables the subjects consumed. 

Thanks for your suggestion. Please see Table 1 for the consumption of fruit and vegetables in the two groups. We included a dedicated part in the discussion (see lines from 288 to 291).

Limitation

It’s likely that this diabetic population is somewhat special from a typical diabetic population with a wider range of ages, and this limitation should be addressed.

Thanks for your comment. We added this limitation to our study (see lines from 324 to 326)

Reviewer 2 Report

  1. Review the structure of the summary. Describe Brief introduction, objective, method, main results and conclusion. Be brief in writing.
  2. Detail in the introduction the dietary patterns associated with diabetes according to the literature.

    3. Point out knowledge gaps in the introduction.

    4. On line 114-115 explain the SECA 700 and SECA 220 (Seca GmBH and Co., Hamburg, Germany).

5. If the study is cross-sectional, why not use the association measure "prevalence ratio"?

Author Response

Firstly, we would like to thank the reviewer for having reviewed accurately our paper and for the helpful advice given.

  1. Review the structure of the summary. Describe Brief introduction, objective, method, main results and conclusion. Be brief in writing.

Thanks for your suggestion. We rephrased the summary by briefly describing the introduction, aim, method, results and conclusion (see lines from 22 to 36)

  1. Detail in the introduction the dietary patterns associated with diabetes according to the literature.

Thanks for your suggestion. We included in the introduction information from the literature on dietary patterns and diabetes (see lines from 77 to 83)

  1. Point out knowledge gaps in the introduction.

Thanks for your comment. We pointed out the gaps at the lines from 82 to 86

  1. On line 114-115 explain the SECA 700 and SECA 220 (Seca GmBH and Co., Hamburg, Germany).

The referee is right. The SECA 700 is a mechanical scale and SECA 220 is a stadiometer (see lines 120-121)

  1. If the study is cross-sectional, why not use the association measure "prevalence ratio"?

We didn’t use this association measure because in our study we evaluate several continuous variables. 

Reviewer 3 Report

Description of dietary patterns associated with diabetes in an older population from Southern Italy, using an unsupervised learning approach

Review report

A brief summary

The topic of the article is interesting and the research is well designed with good scientific contribution. The abstract concisely describes the main parts of the article and gives an overall insight into the research. The introduction sufficiently defines the issue, briefly, clearly and thoroughly. The article is understandable and easy to read.

General concept comments

After the first sentence in the Introduction, please describe the DM subtypes (transfer the second section here). After the explanation, continue about the public health problem and the growing epidemic prevalence.

When expressing numerical values, please make sure to pay attention to the number of decimal places. All numbers should be expressed equally.

Almost half of the references are not within the last 5 years. Please make sure that references are not too old, find a replacement or add more newer references.

Specific comments

Line 1, 2 - It is suggested to shorten the title. Suggestion: „Dietary patterns associated with diabetes in an older population from Southern Italy, using an unsupervised learning approach“

Line 23, 24 – please rewrite the sentence to make it clearer. Please leave out the word „little“ (the same applies to the sentence in Line 71)

Line 63 – If website in brackets is a reference, please make it as a number and mention it in References in the end of the article. 

Line 110 – please correct the typo error (28food)

Line 353 – please check the last section „References“ according to Instructions for authors.

Author Response

Firstly, we would like to thank the reviewer for having accurately reviewed our paper and for the helpful advice given.

General concept comments

After the first sentence in the Introduction, please describe the DM subtypes (transfer the second section here). After the explanation, continue about the public health problem and the growing epidemic prevalence.

When expressing numerical values, please make sure to pay attention to the number of decimal places. All numbers should be expressed equally.

Almost half of the references are not within the last 5 years. Please make sure that references are not too old, find a replacement or add more newer references.

The referee is right. Firstly, we described the DM subtypes after the first senrence in the Introduction. Then, we modified the numbers in order to be expressed equally. Furthermore, we added several new references (16, 24, 29, 33, 37).

Specific comments

Line 1, 2 - It is suggested to shorten the title. Suggestion: „Dietary patterns associated with diabetes in an older population from Southern Italy, using an unsupervised learning approach“ 

Thanks for your suggestion. We modified the title as suggested.

Line 23, 24 – please rewrite the sentence to make it clearer. Please leave out the word „little“ (the same applies to the sentence in Line 71)

Thanks for your suggestion. We rewrite these sentences as required, to make them clearer (see lines from 22 to 24, 67-68)

Line 63 – If the website in brackets is a reference, please make it as a number and mention it in References at the end of the article. 

We added the website in the references (6).

Line 110 – please correct the typo error (28food)

We corrected this error (see line 133)

Line 353 – please check the last section „References“ according to Instructions for authors.

References have been listed according to the instructions.